# Metabolomics of Type 2 Diabetes Mellitus in Sprague Dawley Rats—In Search of Potential Metabolic Biomarkers

**DOI:** 10.3390/ijms241512467

**Published:** 2023-08-05

**Authors:** Innocent Siyanda Ndlovu, Selaelo Ivy Tshilwane, Andre Vosloo, Mamohale Chaisi, Samson Mukaratirwa

**Affiliations:** 1School of Life Sciences, University of KwaZulu-Natal, Westville Campus, Durban 4000, South Africa; syandandlovu111@gmail.com (I.S.N.); vosloo@ukzn.ac.za (A.V.); 2Department of Veterinary Tropical Diseases, Faculty of Veterinary Science, University of Pretoria, Pretoria 0110, South Africa; selaelo.tshilwane@up.ac.za (S.I.T.); m.chaisi@sanbi.org.za (M.C.); 3Foundational Biodiversity Science, South African National Biodiversity Institute, Pretoria 0001, South Africa; 4One Health Center for Zoonoses and Tropical Veterinary Medicine, School of Veterinary Medicine, Ross University, Basseterre KN0101, Saint Kitts and Nevis

**Keywords:** type 2 diabetes, metabolomics, biomarkers, metabolic pathways, Sprague Dawley rats

## Abstract

Type 2 diabetes mellitus (T2DM) is an expanding global health concern, closely associated with the epidemic of obesity. Individuals with diabetes are at high risk for microvascular and macrovascular complications, which include retinopathy, neuropathy, and cardiovascular comorbidities. Despite the availability of diagnostic tools for T2DM, approximately 30–60% of people with T2DM in developed countries are never diagnosed or detected. Therefore, there is a strong need for a simpler and more reliable technique for the early detection of T2DM. This study aimed to use a non-targeted metabolomic approach to systematically identify novel biomarkers from the serum samples of T2DM-induced Sprague Dawley (SD) rats using a comprehensive two-dimensional gas chromatography coupled with a time-of-flight mass spectrometry (GCxGC-TOF/MS). Fifty-four male Sprague Dawley rats weighing between 160–180 g were randomly assigned into two experimental groups, namely the type 2 diabetes mellitus group (T2DM) (n = 36) and the non-diabetic control group (n = 18). Results from this study showed that the metabolite signature of the diabetic rats was different from that of the non-diabetic control group. The most significantly upregulated metabolic pathway was aminoacyl-t-RNA biosynthesis. Metabolite changes observed between the diabetic and non-diabetic control group was attributed to the increase in amino acids, such as glycine, L-asparagine, and L-serine. Aromatic amino acids, including L-tyrosine, were associated with the risk of future hyperglycemia and overt diabetes. The identified potential biomarkers depicted a good predictive value of more than 0.8. It was concluded from the results that amino acids that were associated with impaired insulin secretion were prospectively related to an increase in glucose levels. Moreover, amino acids that were associated with impaired insulin secretion were prospectively related to an increase in glucose levels.

## 1. Introduction

Globally, there is a major dilemma of non-communicable diseases, and among the top ones is type 2 diabetes mellitus (T2DM). T2DM is one of the most prevalent and emerging chronic diseases and metabolic disorders in developing countries and is related to the decline in β-cell function and insulin resistance [1]. Furthermore, T2DM is associated with reduced quality of life from pain or discomfort, mobility, anxiety, and depression [2]. In 2017, the International Diabetes Federation (IDF) atlas reported that approximately 420 million of the world’s human population are diabetic [3], and it is estimated that the number of diabetic patients will surpass 700 million by the year 2045 [4]. The estimated increase in patients with T2DM by the year 2045 is expected to come from regions that experience economic transitions from low-income to middle-income levels [5]. The risk factors associated with T2DM include unhealthy eating, high cholesterol, obesity, and sedentary lifestyle as well as genetic factors [6,7]. Other factors that lead to an increase in the epidemic of T2DM include population aging and urbanization [8].

Individuals with T2DM are characterized by elevated peripheral neuropathy, cardiovascular diseases, and diabetic kidney disease (nephropathy) [9]. Moreover, a study conducted by Scott, Toomath [10] found that patients who had T2DM suffered retinopathy, leading to blindness. Therefore, early diagnosis of T2DM in patients is important so that treatment can be administered to ameliorate the effects of the disease. Methods used for the diagnosis of T2DM include the oral glucose test, fasting blood glucose, and glycated hemoglobin (HbA1c) [11,12]. Despite the availability of the methods, approximately 30–60% of people with T2DM in developed countries are never diagnosed or detected [11]. Some of the reasons include a lack of awareness or misdiagnosis of the disease and a lack of sensitivity of the assays in predicting the diabetic threshold values [13]. Accordingly, there is a strong need for a simpler and more reliable technique for the early detection of T2DM. Additionally, there is a need for new enhanced diagnostic techniques and the application of novel emerging technologies to study T2DM. It has become evident over the last decade that metabolomic techniques and technologies can play an essential role in providing insight into the metabolite changes and pathophysiological pathways of T2DM in humans [14,15].

Metabolomics is a comprehensive systemic study of metabolites in a biological system [16]. For the most part, metabolomics plays a significant role in an in-depth investigation of metabolic pathways and the identification of biomarkers [17]. According to Roberts, Koulman [18], metabolomics is essential for the prediction and early detection of T2DM, and, recently, it has been used to study metabolite changes in patients with T2DM. Furthermore, Ma, Li [4], postulated that the development of metabolomic technology plays an essential role in the early detection and prediction of T2DM. Currently, blood glucose and hemoglobin A1c are two diagnostic biomarkers that exist for the detection of T2DM [19].

Over the past years, different biomarkers have been associated with T2DM [20]. Amino acids, such as tryptophan, leucine, and valine, have been proposed to be useful diagnostic biomarkers for T2DM since the metabolism of these metabolites changes in pre-diabetic patients and over the course of T2DM [15]. Additionally, these identified metabolic biomarkers reflect different metabolic pathways that may be involved in the pathogenesis of T2DM. The application of metabolomics makes it possible to identify the metabolites that can be used as biomarkers for diagnosis, treatment, and prognosis of T2DM [15].

In view of the high incidence of T2DM and its consequences, there is a justifiable need to identify novel biomarkers for T2DM and a need for independent understanding of the metabolic pathways impacted during T2DM. There is still a lack of dependable biomarkers indicative of metabolic changes in T2DM, highlighting the need for markers for early diagnosis and prognosis for T2DM.

Therefore, this study used a non-targeted metabolomics approach to systematically identify novel biomarkers from serum samples of T2DM-induced Sprague Dawley (SD) rats using a comprehensive two-dimensional gas chromatography coupled with a time-of-flight mass spectrometry (GCxGC-TOF/MS).

## 2. Results

### 2.1. Multivariate Statistical Analysis

A total of 941 metabolites were identified in the serum samples from the study animals. When comparing the control with the T2DM group, the unsupervised (PCA), using all 941 metabolites, showed that there was a slight degree of separation between the T2DM and control groups (Figure 1A). The variance captured by the first and the second principal components (PC1 and PC2) were 11.2% and 7.5%, respectively. However, to maximize the separation obtained from the PCA, a PLS-DA was subsequently used (Figure 1B). The performance measurements for the PLS-DA had an accuracy of 0.87, R2 > 0.92, and Q2 > 0.42. Additionally, the distinction between the non-diabetic control and T2DM group was further evaluated using a OPLS-DA plot, constructed using the 941 metabolites, and the plot depicted a clear separation Figure 1C.

### 2.2. Screening of Differential Metabolites

Metabolites that had a significant change (*p* < 0.05) were identified based on the absolute cut-off value of the correlation coefficient and VIP value ≥ 1.5 when performing the PLS-DA, and these were considered potential biomarkers. Of the 941 metabolites in the cleaned dataset, a total of 71 metabolites were determined as metabolites most contributing to the separation between the T2DM and the control group (Appendix A). Considering that the metabolites having a VIP ≥ 1.5 were interpreted as highly influential, the top 15 metabolites were considered to be the most significant potential biomarkers (Figure 2). The VIP plot illustrates the differences in the concentrations of all the potential biomarkers between the two groups. The significant potential biomarkers included six metabolite groups, namely amino acids, carbohydrates, xenobiotics, organic compounds, fatty acids, and glycosides (Table 1). The metabolite aucubin was upregulated in the T2DM group as compared with the control group, with a fold change of 15.22. The five carbohydrate biomarkers identified, namely d-glucose, d-ribofuranose, d-mannitol, and d-galactose, were all upregulated in the T2DM group as compared with the non-diabetic control group. Additionally, these carbohydrate metabolites in the T2DM group had a VIP value > 2, indicating a significant contribution in separating the control from the T2DM group.

A heatmap (Figure 3) highlights some of the metabolites depicting the largest changes between the non-diabetic control and T2DM group. In comparing the two groups, two major hierarchical clusters were observed. Additionally, excluding Methoxy-propanol, H-Imidazole, L-Hydroxy proline, and oxo-butyric acid, the T2DM group had metabolites with higher concentrations/enrichment as compared with the control group.

To confirm further the significance and specificity of the potential discriminating metabolic biomarkers identified from the PLS-DA between the T2DM and the control group, the data were also represented in the form of a volcano plot. The metabolites with a significant change are depicted with red and blue dots, positioned in the upper right and left conner of the volcano plot (Figure 4). The generated volcano plot (FDR-adjusted *p* < 0.05; FC > 2) is depicted in Figure 4, and additional metabolites were discovered; these included L-serine, Furo-pyridine, diethanolamine, phenobarbital, and glycolic acid.

### 2.3. Weekly Changes of the Potential Biomarkers

Changes in the relative concentration of the identified potential biomarkers were observed with the progression of T2DM (Appendix A). From day 0 to day 35, there was a fluctuation in the relative concentration of the identified potential biomarkers. The observed fluctuations within the top 15 potential biomarkers depicted no clear change within the 35 days after diabetes induction. The changes between days showed a significant difference in the D-ribfuranose concentration between day 7 and day 35 (*p* = 0.016) and between day 14 and 35 (*p* = 0.007). Furthermore, the aucubin concentration was significantly different between day 7 and 35 (*p* = 0.015) and between day 14 and 35 (*p* = 0.003). The glucose and L-hydroxyproline concentrations were significantly different between day 14 and day 35 (*p* = 0.034, *p* = 0.014), respectively. The L-hydroxyproline concentration was also significantly different between day 7 and 35 (*p* = 0.040).

### 2.4. Metabolic Pathway Discovery and Analyses of Differential Metabolites

Using the KEGG annotation analysis, the study was able to reveal the metabolic pathways involving the differential metabolites. To explore whether the metabolic pathways were closely related to the experimental conditions, it was important to further analyze the metabolic pathways of the potential metabolites. Metabolite set enrichment analysis (MSEA) was used to identify biologically meaningful patterns that were significantly enriched in quantitative metabolomic data (Figure 5). Metabolic data from the diabetic and control animals based on all the identified biomarkers depicted metabolic pathways that were significantly enriched (FDR < 0.05, *p* < 0.05), and these included ammonia recycling, carnitine synthesis, biotin metabolism, and phenylacetate metabolism. Metabolic pathways that were least enriched in the T2DM serum samples were the urea cycle, selenoamino acid metabolism, and phenylalanine and tyrosine metabolism. Figure 6 represents the highly altered metabolic pathways based on a topology analysis. The pathway impact analysis (*p* < 0.05 and impact value > 0.1) indicated 31 potential metabolic pathways that were impacted during diabetes (Appendix A).

Using the pathway analysis (MetPA) plot to identify the most relevant pathways in the diabetic group, based on the 71 potential metabolic makers, the most significantly affected metabolic pathways, in decreasing order, were aminoacyl-t-RNA biosynthesis; valine; leucine; isoleucine biosynthesis; glyoxylate and dicarboxylate metabolism; phenylalanine, tyrosine, and tryptophan biosynthesis; and alanine, aspartate, and glutamate metabolism (Figure 6). Furthermore, the metabolic pathways most impacted (impact factor > 0.1) included phenylalanine, tyrosine, and tryptophan biosynthesis and glycine, serine, and threonine metabolism, with impact values of 0.5 and 0.46, respectively.

There were eight biomarkers involved in the aminoacyl-t-RNA biosynthesis metabolic pathway, which included the amino acids L-asparagine, L-glutamine, glycine, L-serine, L-leucine-valine, and L-tyrosine. From the KEGG pathway, the differential metabolites between the diabetic group and the control group were involved in multiple metabolic pathways linked to the synthesis of amino acids. Appendix A illustrates the location of the identified biomarkers that were detected on the KEGG map.

### 2.5. Potential Biomarker Verification

The present study also aimed to evaluate the diagnostic utility of serum metabolic profile estimated by GC/GC-TOF-MS for discriminating diabetic patients from the control group. To characterize the predictive value of the identified metabolites independently, a receiver operating characteristic (ROC) analysis was performed (Figure 7). Setting the criteria from the top 15 biomarkers for diagnostic value, 0.8 < AUC < 0.9 was considered good, and 0.9 < AUC ≤ 1.0 was considered excellent. Aucubin and d-galactose 2,3,4,5,6-pentakis showed excellent diagnostic potential for T2DM, having an AUC of 0.967 and 0.956, respectively. Moreover, excluding methyl-heptadecanoic acid, having a moderate diagnostic potential from the top 15 potential biomarkers, the metabolites depicted a good predictive value of more than 0.8. An algorithm combining the 15 most discriminatory metabolites was even more successful, with AUCs > 0.90 (Figure 8).

## 3. Discussion

Using comprehensive metabolomics, this study has identified novel metabolic markers, which included different classes such as carbohydrates, fatty acids, amino acids, xenobiotics, and organic compounds. Additionally, the serum concentrations of the identified markers were significantly different between the T2DM and the non-diabetic control groups. The identified metabolites were found to be involved in multiple pathways, which included ammonia recycling, carnitine synthesis, amino sugar metabolism, aspartate metabolism, and galactose metabolism.

Aucubin was the metabolite marker that was substantially changed. Aucubin is a glycoside that is found in many natural medicines [21]. Additionally, researchers have shown that aucubin has pharmacological properties, including anti-inflammation, anti-aging, anti-tumor, antioxidation, hepatoprotection, neuroprotection, and anti-fibrosis [8]. In the T2DM group, our study showed that aucubin was highly upregulated as compared with the non-diabetic control animals. A study conducted by Jin et al. (2008) [22] looking at the effect of aucubin on the antioxidant status of experimentally induced diabetic rats showed that diabetic rats had a substantial decrease in body weight as compared with the rats that were diabetic and given aucubin. Moreover, Jin et al. (2008) [22] found that streptozotocin-induced diabetic rats had higher blood glucose levels and a decrease in insulin immunoreactivity as compared with diabetic rats treated with aucubin, who had lowered blood glucose levels and a significant increase in insulin immunoreactivity. The study by Jin et al. (2008) [22] also reported that aucubin could stimulate glucose uptake into cells and, therefore, result in a decrease in blood glucose. Therefore, in this study, it can be speculated that the observed upregulation of aucubin was to stimulate glucose uptake into cells of the diabetic rats compared with the control animals. According to Pari and Murugan (2005) [23], the protective function of aucubin in diabetic mice may result from its direct impact on the endocrine pancreas function to make insulin. A common feature in T2DM individuals includes low-grade inflammation [24]. With the mentioned pharmacological properties of aucubin, the increase in the concentration of this metabolite in diabetic rats can be attributed to its anti-inflammation function, and they can be regarded as important in reducing the risks of type 2 diabetes.

After aucubin and glucose, D-ribose was a metabolite that was significantly different and was upregulated in the T2DM group compared was the non-diabetic control group. According to Su and He (2014) [25], D-ribose has been overlooked as a potential risk player in causing the onset of diabetes. A study conducted by Tao et al. (2013) [26] reported an abnormally high level of D-ribose in the urine of diabetic individuals and that the intravenous administration of D-ribose causes a decrease in blood glucose [27]. A study by Su and He (2014) [25] had previously hypothesized that dysfunction or impairment of D-ribose metabolism may have played a role involving the development of complications of T2DM and stated that D-ribose could be used as a biomarker for T2DM. In the present study, D-ribose was part of the potential biomarkers and was associated with the pentose phosphate pathway. The pentose phosphate pathway, also known as the pentose phosphate shunt, is a branch of glycolysis and is essential for the synthesis of ribonucleotides and an important source of Nicotinamide Adenine Dinucleotide Phosphate (NADPH) [28]. According to Ge et al. (2020) [29], pentose phosphate serves as a new and promising target for modulating insulin resistance and obesity-induced inflammation in various tissues. Obesity-induced inflammation can lead to insulin resistance in the liver or skeletal muscles, resulting in systemic insulin resistance [30]. Su and He (2014) [25] reported that D-ribose can be easily measured in the urine of diabetic patients, and it can be assumed that this could be new biomarker for T2DM. According to Cho et al. (2018) [31], the pentose phosphate pathway is interconnected to the glycolysis pathway via the shared use of the three intermediates, namely glyceraldehyde 3-phosphate, glucose 6-phosphate, and fructose 6-phosphate. In this study, with the upregulation or increased glucose concentration observed in the diabetic animals, the pentose phosphate pathway was favored, as this pathway was initiated by glucose that is converted to glucose-6-phosphate (G6P) by hexokinases.

Another carbohydrate that was identified as a potential biomarker for diabetes in our study included D-galactose. According to Gannon et al. (2001) [32], D-galactose can result in increased plasma glucose levels. In this study, D-galactose was found to be upregulated in the T2DM animals in comparison with the non-diabetic control group. Our study findings are in accordance with the study conducted by Hanafy et al. (2021) [33] that reported an increased concentration of D-galactose in T2DM-STZ-induced Wistar albino rats. Additionally, our study also performed a biomarker analysis with universal receiver operating characteristics (ROC) and found that D-galactose had high area under the curve (AUC) scores and high sensitivity and specificity. The AUCs obtained in our study were similar to those of Hanafy et al. (2021) [33]. In our study, D-galactose was involved in different pathways, and this included lactose degradation, galactose metabolism, and sphingolipids metabolism. The upregulation of D-galactose favored these metabolic pathways. Prior to the start of the glycolysis pathway, D-galactose is essential for producing glucose that will initiate the glycolysis pathway. Galactose is essential for cellular metabolism, as it plays an important role in energy production, glycosylation, and storage [34].

Considering other different metabolic pathways affected by T2DM, in the ammonia recycling pathway, four amino acid metabolites were involved: glycine, L-asparagine, L-serine, and L-glutamine. According to Vangipurapu et al. (2019) [35], amino acids are associated with insulin resistance and the risks of T2DM. Moreover, Vangipurapu et al. (2019) [35] found that serine, glycine, and glutamine were associated with improved insulin sensitivity. Amino acids can regulate the secretion of insulin from β-cell lines and primary islet β-cells [22,36]. In the present study, it was found that glutamine was upregulated in the diabetic animals in comparison with the control animals. L-glutamine, the most abundant amino acid in the body, plays an important role in maintaining the function of various cells and organs, such as the heart, kidney, liver neurons, macrophages, pancreatic beta-cells, and intestines [37]. Additionally, glutamine plays an essential role in the production of glutathione, which is important for reducing oxidative stress, which eventually maintains the inflammatory processes in the β-cells in diabetic individuals [38]. Studies conducted on the possible effect of glutamine on the glycemic status of diabetic animals reported that glutamine supplementation caused a reduction in the plasma glucose levels, with an increase in the pancreatic and plasma levels of insulin [39,40]. In diabetic patients, hyperglycemia is one of the symptoms of T2DM and later results in complications of the disease since an increase in glucose level induces peroxidation and directly injures cells [41].

Other glucogenic metabolites that were detected in the present study included glycine, serine, and L-valine. Previous research suggests that changes in plasma glycine may be one of the prediabetes biomarkers [42]. These metabolites were involved in different metabolic pathways. In this study, glycine was another amino acid that was significantly upregulated in the diabetic animals in comparison with the control animals. This metabolite was associated with ammonia recycling, carnitine synthesis, methionine metabolism, alanine metabolism, purine metabolism, and porphyrin metabolism. A study conducted by Chen et al. (2018) [43] looking at the effect of glycine in diabetic rats reported that rats that were diabetic and given glycine had higher insulin secretion compared with those that were diabetic and went without glycine. Additionally, Chen et al. (2018) [43] reported that the administration of glycine improved the β-cell microstructure, increased the β-cell mass, and increased the number of mature insulin-secreting particles. Another study conducted by Li et al. (2019) [44] reported that glycine increased the glutathione content, reduced blood glucose levels, and increased body weight. Floegel et al. (2013) [45] also reported that an increase in glycine concentration was associated with elevated insulin sensitivity and secretion. Therefore, the observed increase in glycine in this study in the diabetic rats can be attributed to the role that glycine plays in diabetic individuals, which includes being an antioxidant, as oxidative stress is considered to be the main pathogenic mechanisms of β-cell damage in diabetes. In our study, having increased glucose in the diabetic animals, it can be speculated that the increase in glycine served to attenuate the glucose levels by increasing the secretion of insulin. In the diabetic rats, the glucose concentration was reported to be significantly reduced by supplemental glycine [46,47,48]. Therefore, it can be stated that glycine played a significant role in glucose homeostasis and that this metabolite supplementation improved the glucose tolerance.

In this study, the ammonia recycling pathway included metabolites such as L-asparagine, L-tyrosine, and L-serine. The mentioned biomarkers were also linked with aspartate metabolism, phenylalanine, and tyrosine metabolism, methionine metabolism, homocysteine degradation, and thyroid hormone synthesis. A study by Vangipurapu et al. (2019) [35] on the association of amino acids and T2DM reported that L-asparagine and L-serine were associated with improved insulin sensitivity and insulin secretion. However, Vangipurapu et al. (2019) [35] also found that the amino acid L-tyrosine had the largest effect on the reduction of insulin sensitivity and secretion. In the present study, L-tyrosine and asparagine were upregulated in the diabetic animals. According to Würtz et al. (2013) [15], aromatic amino acids, including L-tyrosine, were associated with the risk of future hyperglycemia and overt diabetes. In patients who have a diabetic foot ulcer, the concentration of L-tyrosine was highly elevated [49]. In the present study, the metabolic pathway with which these biomarkers were associated were pathways that played a significant role in the production of amino acids. The increased concentration of L-serine in diabetic animals favored the methionine metabolism, which is a pathway that can regulate metabolic processes, digestive functioning in mammals, and the innate immune system [50]. Moreover, according to Ribas et al. (2014) [51], an increase in methionine affected insulin resistance and could predict the risks of developing diabetes. According to Han et al. (2018) [52], homocysteine was shown to be an intermediate in methionine metabolism. Additionally, elevated homocysteine levels were linked with the early development of heart diseases, especially pronounced in patients with diabetes. In our study, it was found that the homocysteine metabolism pathway was impacted by diabetes, and L-serine was the metabolite involved. Recent systematic reviews and studies have concluded that high concentrations of homocysteine may be associated with the progression of T2DM [53].

## 4. Methods and Materials

### 4.1. Experimental Design

Fifty-four male Sprague Dawley rats weighing between 160–180 g were randomly assigned to two experimental groups, namely the type 2 diabetes mellitus group (T2DM) (n = 36) and the non-diabetic control group (n = 18) (Figure 9). Experimental rats were bred and maintained at the Biomedical Resource Unit (BRU), University of KwaZulu-Natal, Westville Campus in Durban, South Africa. They were maintained and subjected to laboratory conditions of constant room temperature (22 ± 2 °C), carbon dioxide content of <5000 ppm, 12 h light/dark cycle, and relative humidity of 55 ± 5%. They were fed with standard rat chow diet (Meadows feeds, Pietermaritzburg, South Africa), and water was given ad libitum daily throughout the experimental period. Rats from both groups were euthanized using isoform inhalation in a gas chamber at days 0, 7, 14, 21, 28, and 35 after induction of diabetes, and on each day of sacrifice, six rats were euthanized from the T2DM group, while three rats were euthanized from the control group (Figure 9).

### 4.2. Induction of Type 2 Diabetes Mellitus

Type 2 diabetes mellitus was induced in the experimental rats as described by [54] and Islam and Wilson [55]. To induce insulin resistance, rats in the T2DM group were given 10% fructose solution orally ad libitum for 14 days, and those in the control group received normal water. After 14 days of taking fructose solution, T2DM group was fasted overnight, and streptozotocin (STZ) (Sigma, St. Louis, MO, USA) was administered interperitoneally at a dose of 40 mg/kg bw. STZ solution used was first freshly dissolved in citrate buffer (pH 4.5) before being administered. Thereafter, blood from the tail vein was collected the following morning to determine the non-fasting blood glucose (NFBG) level using a glucometer (Glucoplus Inc., Saint-Laurent, QC, Canada). Rats that had glucose levels > 18 mmol/L were considered diabetic, and those that had glucose levels < 18 mmol/L were considered non-diabetic and were excluded from the study. Thereafter, the glucose level in diabetic rats was measured every two days throughout the experiment.

### 4.3. Terminal Studies

Rats were sacrificed from groups at days 0, 7, 14, 21, 28, and 35 post-diabetic induction. On each day of sacrifice, six animals from the T2DM group and three animals from the control group were euthanized using ISOFOR inhalation for 3 min in an anesthetic chamber. Cardiac puncture was used to collect blood into 10 mL tubes that contained clotting activator (dya cgel and clot activator tubes, Terumo^®^ Venosafe^®^, Terumo, Tokyo, Japan). Thereafter, the blood was centrifuged (Thermo-scientific Heraeus Labofuge 200 microprocessor, Waltham, MA, USA) at 132× *g* for 15 min at a temperature of 4 °C. After centrifuging, serum was collected into Eppendorf micro-centrifuging tubes and stored in a Bio Ultra freezer (Snijers Scientific, Tilburg, The Netherlands) at −80 °C until analysis.

### 4.4. Analysis of Serum Samples

A total of 54 serum samples from blood of individual rats (aliquots of 2 mL) were collected from the T2DM (n = 36) and non-diabetic control (n = 18) groups and stored at −80 °C in a Bio Ultra freezer (Snijders Scientific, Tilburg, The Netherlands) until they were transported to Potchefstroom, North-West University, Center for Human Metabolomics for analysis using a GCxGC-TOF/MS. The whole metabolome was extracted from the serum samples by applying the protein crush method. Briefly, serum samples were aliquoted in 1.5 mL Eppendorf tubes, 50 μL of the internal standard (3-phenyl butyric acid 50 ppm) was added to 50 μL of each serum sample, and 300 μL ice-cold acetonitrile was added, vortexed, and incubated on ice for 10 min. After incubation, the sample mixtures were centrifuged at 13,000 rpm for 10 min at 4 °C. The supernatant was transferred into a gas chromatography vial and dried under a gentle stream of nitrogen at 40 °C for 20–30 min. For derivatization purposes, 50 μL methoxyamine HCl (150 mg in 10 mL pyridine) was added, followed by an incubation step at 50 °C for 90 min. Thereafter, 40 μL BSTFA + 1% TMCS was added, and the extract was re-incubated at 50 °C for 60 min. The extracts were then transferred to a 250 μL insert in a sample vial and capped before GCxGC-TOF-MS analysis.

### 4.5. Untargeted GCxGC-TOFMS Approach

#### 4.5.1. GCxGC-TOFMS Analysis

Pegasus GCxGC-TOFMS (Leco Corporation Joseph, MI, USA) that uses an Agilent 7890A GC (Atlanta, GA, USA) coupled to a time-of-flight mass spectrometer (TOFMS) (Leco Corporation, St. Joseph, MI, USA) equipped with a Gerstel Multipurpose sampler was used for chromatographic analyses of the derivatized samples. Here, 1 µL of serum extract was randomly injected at a split ratio of 1:50, and the carrier gas used was helium at a flow rate of 1 mL min^−1^. For the entire run, the temperature of the injectors was kept constant at 270 °C. A Restek Rxi-5Sil MS capillary column (29.145 m × 0.25 µm d.f.) (Corporation, Bellefonte, PA, USA) was used as the primary column. The primary oven was programmed to an initial temperature of 70 °C for 2 min to obtain a compound separation. Subsequently, this was followed by a 4 °C per minute increase to a final temperature of 300 °C, which was maintained for 2 min. The second separation of compounds was achieved using a Restek Rxi-17 (Corporation, Bellefonte, PA, USA) (1.400 m, 0.25 µm i.d., 0.25 µm d.f.) column, and the secondary column was set to the same temperature parameters as that of the primary column. The filament bias was EI at 70 eV, while the detector voltage was at 1600 V. Subsequently, the mass spectra were collected at an acquisition rate of 200 spectra per second, with a source temperature of 220 °C and a solvent delay of 400 s from 50 to 800 *m*/*z* [56].

#### 4.5.2. Peak Identification

Leco Corporation Chroma-TOF software (version 4.50) was used to obtain peak finding and mass spectral deconvolution at an S/N ratio of 100, with a minimum of 3 apexing peaks. Using the mass fragmentation patterns generated by the MS, together with their respective GC retention times, the identities of these peaks were determined by comparing them with commercially available NIST spectral libraries (mainlib and replib).

### 4.6. Data Clean-Up

Data clean-up was conducted using Microsoft Excel to remove features that were not reliably measured. The reliability of each variable was assessed by calculating the relative standard deviation (standard deviation divided by the mean) across all quality control samples of T2DM and control samples. Features with a relative standard deviation above 50% were excluded from further analysis. This was accomplished by replacing all zero values with half the minimum observed value for the dataset, which served as an estimate for the detection limit. The data were log-transformed and scaled using MetaboAnalyst for data that were not normally distributed. Natural-shifted log transformation was performed to correct the skewness distribution of the variables, followed by auto-scaling to replace all variables to obtain normality [57].

### 4.7. Metabolic Biomarker Discovery and Pathway Analysis

Identification of the potential biomarkers was based on the greatest variable importance in the projection (VIP) value and had to be statistically significant (VIP > 1.5 and *p* value < 0.05) [58]. The identified metabolic biomarkers were mapped onto the Kyoto Encyclopedia of Genes and Genome (KEGG) pathway network using iPath 3.0 (http://pathways.embl.de, accessed on 22 July 2023), a web-based application used to visualize and analyze cellular pathways. A metabolite set enrichment overview (MSEA) was constructed using the significant metabolites to elucidate the pathways in which these significant metabolites were involved by using MetaboAnalyst version 5.0 (https://www.metaboanalyst.ca/docs/Publications.xhtml, accessed on 22 July 2023), an online tool for analyzing and interpreting metabolic enrichment. Moreover, metabolic pathway analysis (MetPA) was incorporated with MetaboAnalyst for pathway analyses.

### 4.8. Statistical Analysis

Data were analyzed by using MetaboAnalyst version 5.0 and SPSS Version 28.0, and the graphical representation of data was created on Prism software version 5. The integral intensity of the metabolites was presented as median with interquartile range (IQR). Prior to principal component analysis (PCA), data were normalized using auto-scaling and log transformation. For data overview and pattern discovery, a PCA was first constructed followed by a supervised classification method, partial least-squares discriminant analysis (PLS-DA). Thereafter, a supervised orthogonal partial least-squares discriminant analysis (OPLS-DA) method was performed to improve the separation between the groups of samples and to minimize other biological analytical variations. The OPLS-DA was cross-validated by permutation (n = 20). Thereafter, the goodness-of-fit parameters were calculated (R2X, R2Y, and Q2Y). The data were also represented in the form of a volcano plot to distinguish metabolic biomarkers between the T2DM and control groups. Analysis of variance (ANOVA) was applied to the potential biomarkers to determine whether there were statistically significant differences in the metabolite concentrations among weeks. The receiver operating curve analysis (ROC) was used to further evaluate the predictive ability of the identified potential metabolic biomarkers. Moreover, the area under the curve was used to determine the diagnostic accuracy of the biomarkers, where 0.8 < AUC < 0.9 was considered good, and 0.9 < AUC ≤ 1.0 was considered excellent.

## 5. Conclusions and Recommendations

All amino acids identified were associated with a higher risk of T2DM, whereas glycine and glutamine were associated with a lower risk. Amino acids that were associated with impaired insulin secretion were prospectively related to an increase in glucose levels. Our study results showed that the changed glucogenic amino acids, such as glycine, were associated with the impairment of insulin sensitivity at a later phase of diabetes progression as compared with branched and aromatic amino acid metabolism. The GCxGC-TOF-MS platform proved effective in screening the serum samples of diabetic animals, as it identified that disturbances in various metabolic pathways translated into significantly altered metabolites. The ROC curve analysis allowed us to obtain a general overview of the significant alterations of biomarkers in the T2DM patients with complications. The limitation of the study is that it did not investigate how the altered concentrations of different amino acids affected the insulin resistance in the rats. As this was a lab-based study, in human studies, it can be recommended to target the specified potential amino acids biomarkers based on their impact on insulin resistance and sensitivity. A better understanding of these will aid in a better understanding of T2DM.

## Figures and Tables

**Figure 1 ijms-24-12467-f001:**
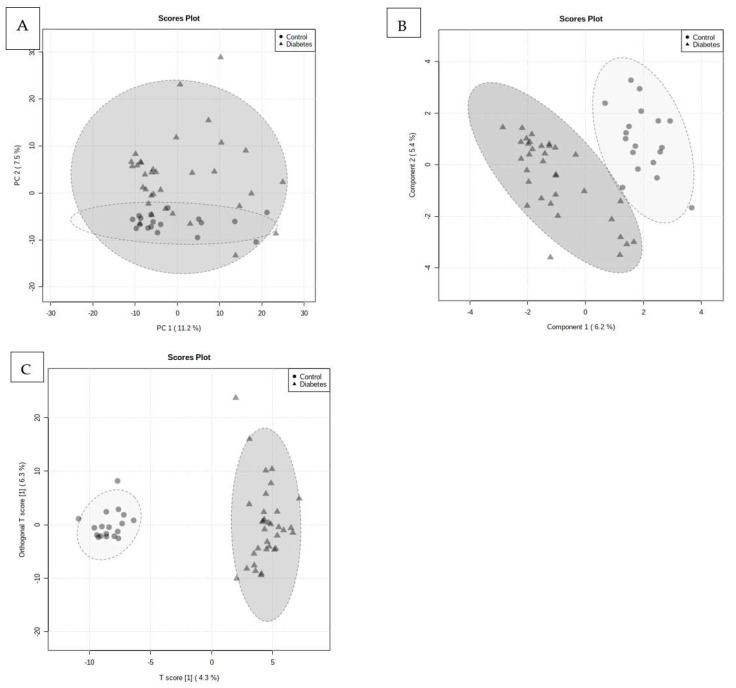
PCA, PLS-DA, and OPLSDA score scatter plots distinguishing between the T2DM and control group. (**A**) PCA (PC1 and PC2: 11.2% and 7.5% of the total variance, respectively); (**B**) PLSDA (PC1 and PC2: 6.2% and 5.4% of the total variance, respectively); (**C**) OPLSDA (obtained with the predictive and orthogonal component, accounting for 4.3% and 6.3% of the total variance, respectively) between the T2DM and the non-diabetic control group, indicating that these groups had different metabolite profiles. The goodness-of-fit of the parameters (OPLS-DA, R2X, R2Y, and Q2Y) for the OPLS-DA was calculated. R2X and R2Y represented the fraction of variance of the X and Y variables of the model, while Q2Y represented the predictive performance of the model. The values of R2Y and Q2Y of the model were 0.97 and 0.51, respectively.

**Figure 2 ijms-24-12467-f002:**
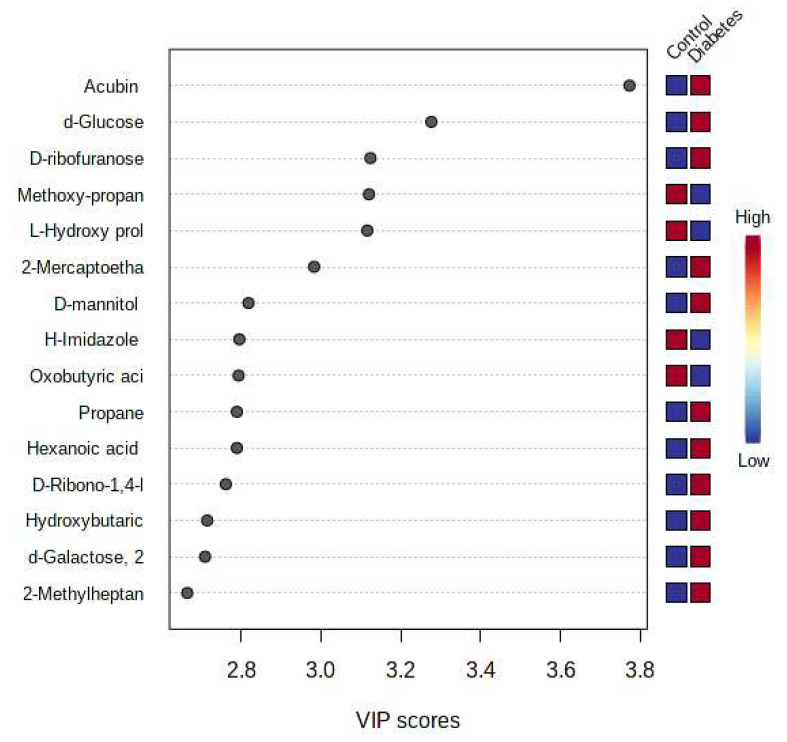
Metabolites ranked by variable importance in projection (VIP) scores for the type 2 diabetes mellitus (T2DM) and non-diabetic control group. On the right, colored boxes indicate the relative concentration of the metabolite in the two groups under study.

**Figure 3 ijms-24-12467-f003:**
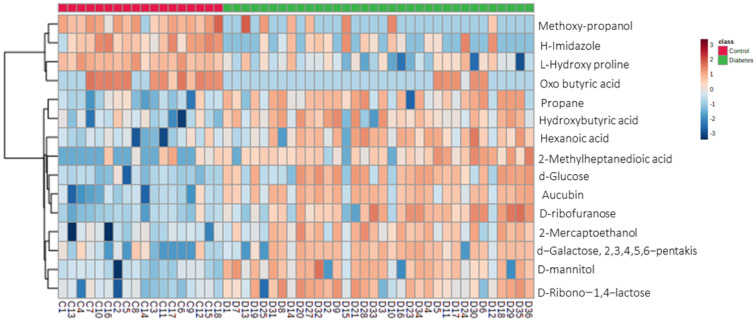
Heatmap visualization of group averages based on 15 top potential biomarkers. Rows: potential biomarkers; columns: samples. Red: control group; green: T2DM. Color keys indicate metabolite values (dark blue: lowest; dark red: highest).

**Figure 4 ijms-24-12467-f004:**
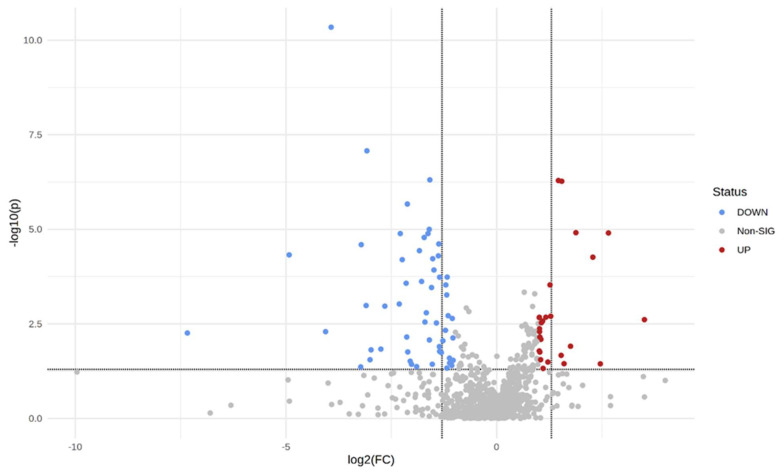
Volcano plot. Each dot represents a metabolite. The abscissa represents the fold change values, and the ordinate depicts the *p*-value of the Student’s *t*-test.

**Figure 5 ijms-24-12467-f005:**
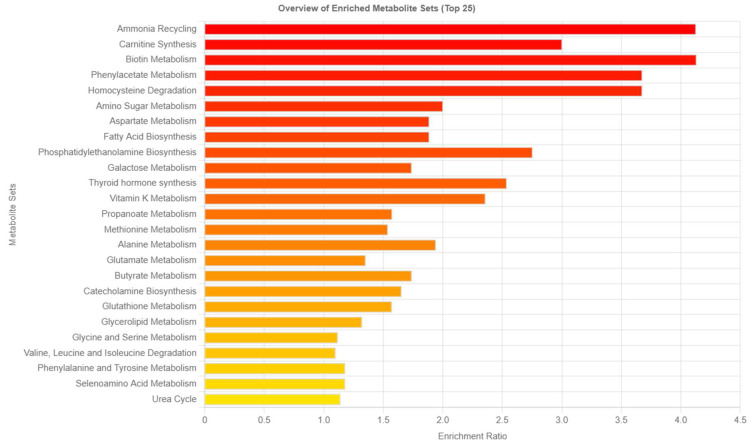
Metabolic pathways associated with 71 identified potential metabolites. The horizontal bars show a summary of metabolic pathways that were strongly affected or changed in the T2DM SD rats compared with the control group. Pathways are shown in order of decreasing significance from top to bottom (increasing nominal *p*-values, colored from red to yellow), with bars indicating their estimated fold enrichment.

**Figure 6 ijms-24-12467-f006:**
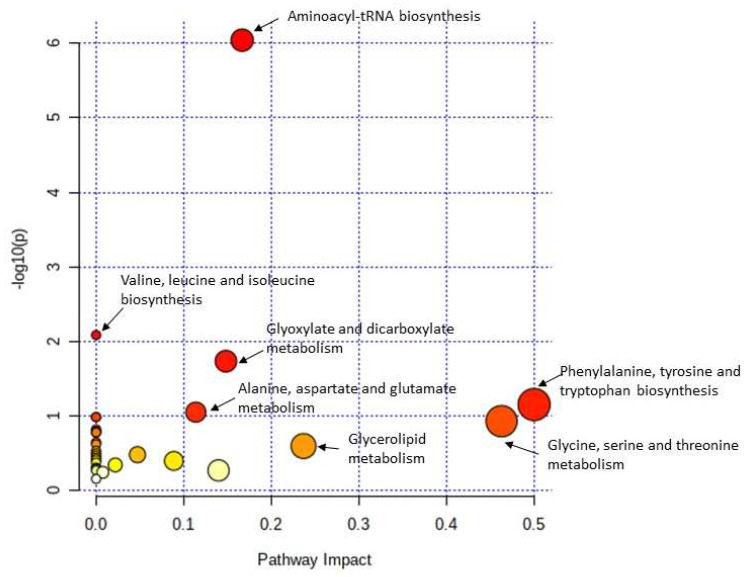
Pathway enrichment versus −log10 (*p*) with *p* < 0.01 cut-off for comparison of diabetic rats and controls. Metabolic pathways associated with the top 71 identified metabolite markers. All the matched pathways are displayed as circles. The node size is proportional to the enrichment ratio. Light yellow to red indicates the *p*-values from small to large. The color and size of each circle are based on the *p*-value and pathway impact value, respectively. The most impacted pathways having high statistical significance scores are indicated by their names.

**Figure 7 ijms-24-12467-f007:**
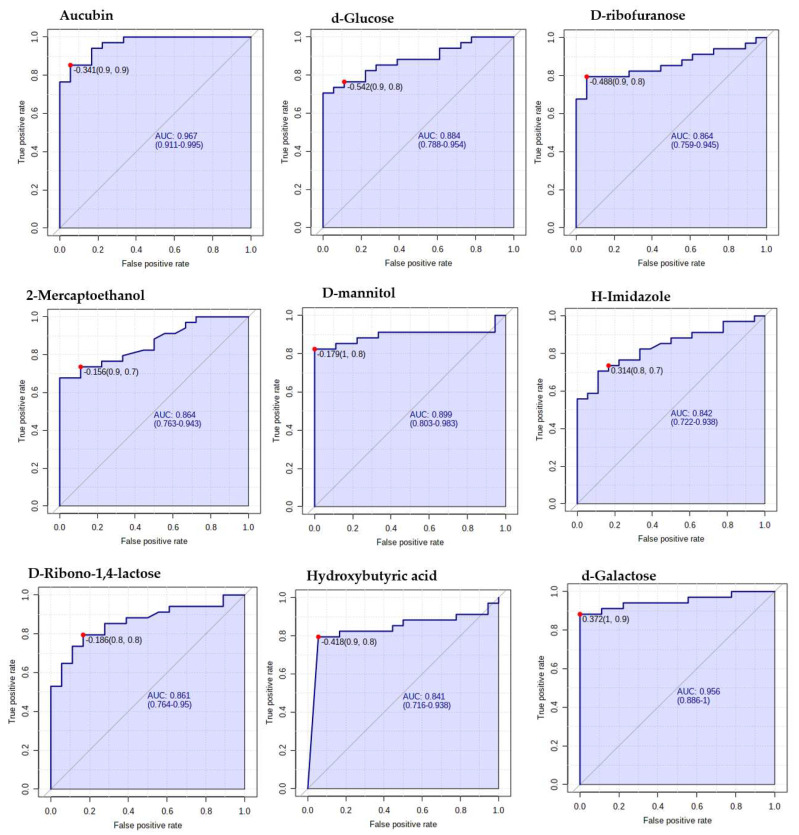
Verification of the top 15 identified potential biomarkers. ROC analysis to depict further the predictive value of these individual metabolites independently.

**Figure 8 ijms-24-12467-f008:**
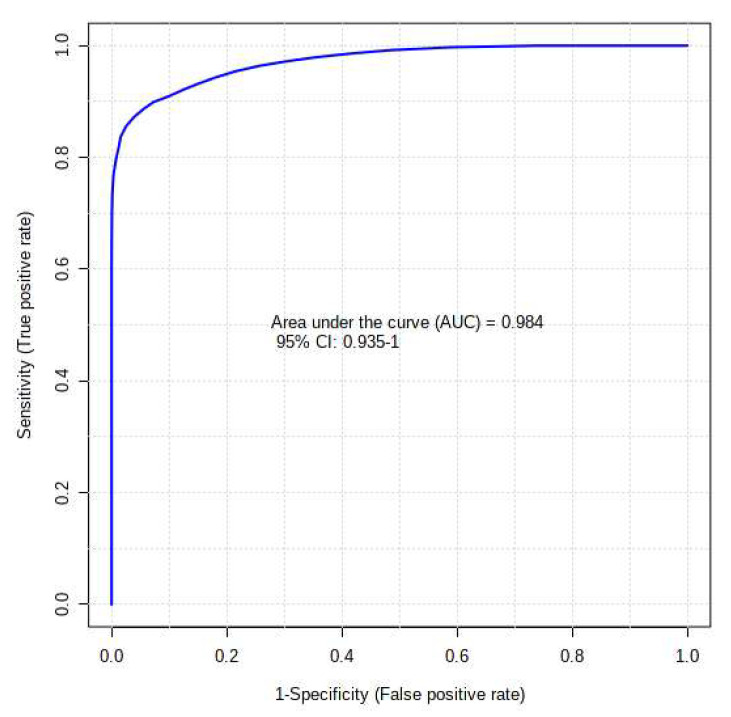
The receiver operating curve of the top 15 identified potential biomarkers for distinguishing the control group and T2DM group.

**Figure 9 ijms-24-12467-f009:**
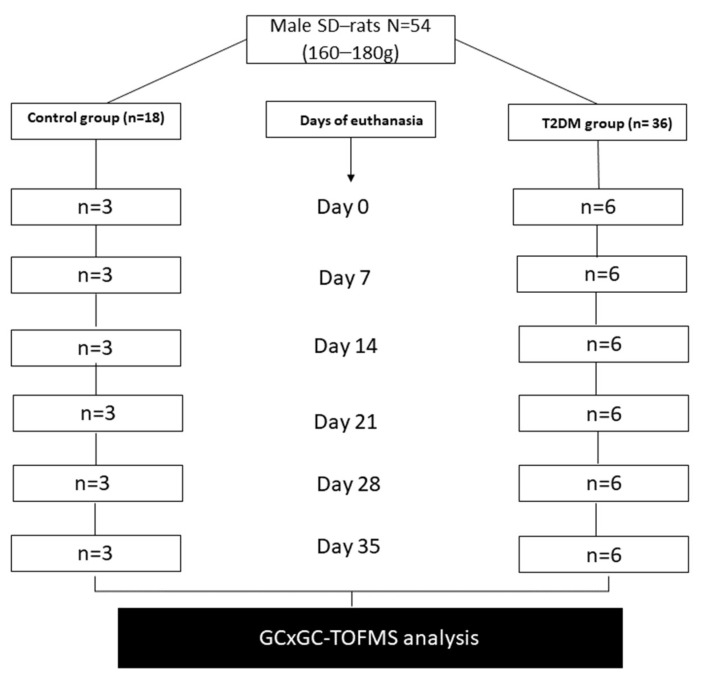
A schematic diagram of the experimental design.

**Table 1 ijms-24-12467-t001:** Quantitative comparison of the top 15 potential biomarkers from the serum samples of T2DM-induced Sprague Dawley rats and control group.

Names	Type/Class	VIP	FC	Regulation	*p*-Value
Aucubin	Glycoside	3.773	15.224	Up	4.55 × 10^−11^
d-Glucose	Carbohydrate	3.276	8.451	Up	8.4 × 10^08^
d-ribofuranose	Carbohydrate	3.124	3.002	Up	4.92 × 10^−07^
Methoxy-propanol	Xenobiotic	3.120	0.362	Down	5.13 × 10^−07^
L-Hydroxy proline	Amino acid	3.116	0.342	Down	5.36 × 10^−07^
2-Mercaptoethanol	Xenobiotic	2.983	0.230	Up	2.15 × 10^−06^
D-mannitol	Carbohydrate	2.819	0.329	Up	1 × 10^−05^
H-Imidazole	Organic compound	2.796	3.681	Down	1.23 × 10^−05^
4-Oxo butyric acid	Fatty acid	2.794	0.159	Down	1.25 × 10^−05^
Propane	Xenobiotic	2.789	4.880	Up	1.29 × 10^−05^
Hexanoic acid	Fatty acid	2.789	3.095	Up	1.3 × 10^−05^
D-Ribono-1,4-lactose	Organic compounds	2.762	3.294	Up	1.64 × 10^−05^
Hydroxybutyric acid	Organic compounds	2.715	2.591	Up	2.44 × 10^−05^
d-Galactose	Carbohydrate	2.710	9.277	Up	2.55 × 10^−05^
2-Methylheptanedioic acid	Organic compound	2.665	3.564	Up	3.67 × 10^−05^

## Data Availability

The original contributions presented in the study are included in the article/Appendix A; further inquiries can be directed to the corresponding author.

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
