# Peer review of "Metabolomics of Type 2 Diabetes Mellitus in Sprague Dawley Rats—In Search of Potential Metabolic Biomarkers"

_ijms, 2023, doi:10.3390/ijms241512467_

Round 1

Reviewer 1 Report

The structure of the research is well organized, and the relevance of the topic selection is significant. The subject is presented in an engaging and coherent manner. The structural and content development of the paper is logical, and the Author expresses their thoughts clearly and unambiguously, using appropriate terminology. The scientific work presents valuable results from a clinical perspective, which can also be applied to everyday practice. The "Results" section is logically structured, statistically interpretable, and facilitates data interpretation with high-quality figures. The "Discussion" section is well-developed, providing sufficient depth and supporting the integration of the work into international research trends.

Comments and question:

1. The manuscript structure is unusual, as "Materials and Methods" typically follows the introduction, not the discussion.

2. It would be worth including limitations in the paper. For example, the experimental data may not necessarily be applicable to human samples. The paper did not investigate how the altered concentrations of different amino acids affect insulin resistance in rats. What impact the altered amino acid pattern may have on the extent of insulin resistance in diabetic rats?

After incorporating the requested modifications, I recommend this excellent work for publication.

Author Response

comments and responses

  1. The manuscript was prepared as per the journals' guidelines and requirements. The methods and materials come after the discussion section.
  2. On line 637-640, the information suggested was outlined which stated the limitations of the study with future steps.  

Reviewer 2 Report

The manuscript is written well and interesting to the readers. All the experiments are explained well. However, I have a few comments which need to be addressed before publication. 

  Why did the author choose only serum samples, not the blood or feces samples?  LC-MS techniques are well known instruments for non volatile and non targeted techniques as compared to 2D GC-MS MS. Why did authors use these techniques?  Why did the author choose 35 days of the study? Is there any reference for this kind of study? It would have been more informative if authors may have used the same mouse for over all studies rather than sacrificing after 0, 7, 14, 21, 28, and 35 post-diabetic induction. 

Minor editing of English language required. 

Author Response

Why did the author choose only serum samples, not the blood or feces samples? 

  • The serum is a common sample of convenience for metabolomics studies. Moreover, serum is widely used in clinical studies and provide higher sensitivity for biomarker discovery studies and metabolite concentrations are reported to be higher than in plasma samples.

LC-MS techniques are well known instruments for nonvolatile and non-targeted techniques as compared to 2D GC-MS MS. Why did authors use these techniques? 

  • In this study, we used the two-dimensional spectrometry, which can cover more metabolites as compared to the one-dimensional spectrometry which resulted us in quantifying more metabolites as compared to using a one dimension. Moreover, the gas phase covers a wider range of global metabolites compared to the liquid phase mass spectrometry.
  • Another reason as to why this study used the gas phase is that it has been proven to be an effective tool for identification and quantification of metabolites in mammalian cell lines due to its high resolution, sensitivity, separation capacity, and its ability to generate a mass spectrum for each compound, allowing separation and detection of many metabolites.
  • Lastly, with the LC–MS method, the presence of non-linear shifts for the retention times on different days and the presence of many peaks arising from isotopes, adducts, degradation products or pollutants could further complicate the processing of raw data. These types of problems are less frequent in analysis performed with the gas phase.

Why did the author choose 35 days of the study?

Is there any reference for this kind of study? It would have been more informative if authors may have used the same mouse for overall studies rather than sacrificing after 0, 7, 14, 21, 28, and 35 post-diabetic induction. 

  • Thank you for raising this point. In the interest of reducing the number of animals in the study, we did consider the reviewer’s proposition. However, the proposed method could not guarantee us to get us enough blood to give us the quantity of serum we needed at each point and hence we resorted to sacrificing a proportion of the animals and also took advantage of collecting organs from the sacrificed animals for a different study related to T2DM altogether. The issue of minimizing stress and to some extend pain to the animal was considered when the proposal was reviewed through the University ethics committee.
  • 35 days were chosen as sufficient time to observe progression of diabetic animals with time and observe if there were developing any adaptation mechanisms and we believe 5 weeks were sufficient and also the cost of laboratory analysis of the samples was taken into account.